# FedHyMoe: Hypernetwork-Driven Mixture-of-Experts for Federated Domain Generalization

## Abstract

Federated Learning (FL) enables collaborative model training without sharing raw data, but most existing solutions implicitly assume that each client's data originate from a single homogeneous domain. In practice, domain shift is pervasive: clients gather data from diverse sources, domains are heterogeneously distributed across clients, and only a subset of clients participate in each round. These factors cause substantial degradation on unseen target domains. Prior Federated Domain Generalization (FedDG) methods often assume complete single-domain datasets per client and sometimes rely on sharing domain-level information, raising privacy concerns and limiting applicability in real-world federations. In this paper, we introduce **FedHyMoe**, a **Hypernetwork-Driven Mixture-of-Experts framework** that addresses these challenges by shifting from parameter-space fusion to embedding-space parameter synthesis. Each client is represented by a compact domain embedding, and a shared hypernetwork generates its Mixture-of-Experts (MoE) adapter parameters. At test time, unseen domains are handled by attending over source client embeddings to form a test-domain embedding, which the hypernetwork uses to synthesize a specialized adapter. This enables non-linear interpolation and extrapolation beyond convex averages of stored parameters, while reducing communication and storage overhead and mitigating privacy risks by exchanging only low-dimensional embeddings. FedHyMoe consistently achieves higher generalization accuracy and improved calibration compared to baselines under domain heterogeneity and partial participation highlighting embedding-driven hypernetwork synthesis as a powerful inductive bias for robust, efficient, and privacy-conscious Federated Domain Generalization.

## 1 Introduction

Deep neural networks (DNNs) thrive on large, centralized datasets (Krizhevsky et al., 2012; He et al., 2016; Dosovitskiy et al., 2020; Liu et al., 2021b), yet real-world data are fragmented across silos where privacy rules forbid sharing. Federated learning (FL) (McMahan et al., 2017a; Li et al., 2020) enables collaborative training without raw data exchange, but faces two core obstacles: (i) non-i.i.d. client distributions that impair convergence (Li et al., 2019; Karimireddy et al., 2020), and (ii) domain shift, where test data differ systematically from training clients (Liu et al., 2021a; Zhang et al., 2023; Bai et al., 2023). This motivates the setting of *federated domain generalization* (FDG): training across decentralized sources to generalize to *unseen target domains* under strict communication and privacy constraints (Zhang et al., 2021; Yuan et al., 2023; Seokeon et al., 2024).

Most FDG methods remain tied to *parameter-space fusion*, where client-specific models or adapters are linearly aggregated (e.g., FedAvg and its variants (McMahan et al., 2017a; Li et al., 2019; Karimireddy et al., 2020; Wang et al., 2020)). While simple, this approach suffers from a *convex-fusion ceiling*: it can only interpolate within the convex hull of source parameters, leaving unseen domains poorly represented (Zhang et al., 2023; Yuan et al., 2023; Bai et al., 2023). Moreover, transmitting large parameter blocks inflates bandwidth requirements and heightens privacy risk (Geiping et al., 2020; Huang et al., 2021; Hatamizadeh et al., 2023).

This paper presents a novel framework for FDG, termed **FedHyMoe**, which leverages a *hypernetwork* as the central generator. A hypernetwork is a neural network that outputs the parameters of another network conditioned on a compact embedding. In FL, hypernetworks have been explored for personalization and label heterogeneity, and more recently for FDG through hypernetwork-based fusion approaches. Parallel advances in FDG also highlight the effectiveness of mixture-of-experts (MoE) fusion strategies (Radwan et al., 2025), while privacy-preserving efforts focus on mitigating gradient inversion attacks via structural indirection or defense mechanisms (Guo et al., 2025). However, prior approaches predominantly generate client-specific models for training or personalization. In contrast, our framework realizes a different goal: transforming the privacy-aligned indirection of hypernetworks into a mechanism for *domain generalization*.

FedHyMoe reframes adaptation as *embedding-space composition* followed by *hypernetwork-based parameter synthesis*. Each client is summarized by a compact *domain embedding*, and a shared hypernetwork maps embeddings into *Mixture-of-Experts (MoE)* adapters with Kronecker/low-rank structure. At test time, a batch of target data produces a *test embedding*, which attends over stored source embeddings to yield similarity weights. These weights pool into a descriptor that the hypernetwork transforms into a specialized adapter enabling parameterization for unseen domains through non-linear synthesis rather than averaging. Such a design strictly generalizes convex fusion, since with one-hot embeddings and a linear generator FedHyMoe reduces exactly to standard parameter averaging (Radwan et al., 2025). The true advantage emerges once the generator leverages non-linear embedding-to-parameter mappings, which enable interpolation and extrapolation beyond the convex span of source models. In this way, FedHyMoe turns the privacy-aligned indirection of hypernetworks into a generalization mechanism: domain embeddings are composed at test time, and parameters are generated to implement the right function for unseen domains. Communication and storage requirements scale only with the embedding dimension rather than the adapter size, with Mixture-of-Experts and Kronecker decompositions providing the right balance of diversity, compactness, and communication efficiency. Finally, privacy exposure is substantially narrowed, as the server observes only SecAgg-protected generator gradients and compact embeddings while never accessing raw domain statistics or full parameter blocks (Guo et al., 2025).

**Contributions.**

- We introduce **FedHyMoe**, an FDG framework that replaces parameter-space averaging with *embedding-space attention* and *hypernetwork-based adapter synthesis*.
- We instantiate *hypernetwork-generated MoE adapters* with Kronecker/low-rank factors, achieving strong accuracy efficiency trade-offs under partial participation.
- We establish that convex adapter fusion is a strict special case of our formulation, and provide inference-only controls that disentangle attention from synthesis.

## 2 RELATED WORK

### 2.1 GRADIENT INVERSION AND PRIVACY IN FEDERATED LEARNING.

Federated learning (FL) mitigates direct exposure of raw data by training models through decentralized updates; however, a substantial body of work on *gradient inversion* and related reconstruction attacks has shown that shared updates (gradients or parameter deltas) can leak sensitive information, including approximate input reconstructions, membership, and attributes (Fredrikson et al., 2015; Zhu et al., 2019; Geiping et al., 2020). These risks intensify in regimes with high–capacity vision backbones and small, skewed client datasets, where updates become more uniquely tied to local samples. This observation motivates defenses that (i) *minimize the exposure surface* by restricting the dimensionality and informativeness of communicated artifacts, and/or (ii) *alter the communication primitive* so that only privacy-preserving aggregates (e.g., masked or securely aggregated sums) are revealed rather than raw per-client updates.

### 2.2 PRIVACY-PRESERVING TECHNIQUES IN FL.

Privacy protection in FL largely follows three methodological lines. (1) *Secure multi-party computation* (SMC) and secure aggregation ensure that only masked aggregates of local updates are

revealed (Yao, 1982; Bonawitz et al., 2017; Mugunthan et al., 2019; Mou et al., 2021). (2) *Homomorphic encryption* (HE) enables computation on encrypted parameters but typically incurs substantial communication and computation costs (Gentry, 2009; Park & Lim, 2022; Ma et al., 2022). (3) *Differential privacy* (DP) clips and perturbs updates to provide formal guarantees, often at the expense of model utility (Geyer et al., 2017; McMahan et al., 2017b; Yu et al., 2020; Bietti et al., 2022; Shen et al., 2023). Additional empirical defenses—gradient pruning/masking and noise injection (Zhu et al., 2019; Huang et al., 2021; Li et al., 2022; Wei et al., 2020)—as well as specialized frameworks such as Soteria, PRECODE, and FedKL (Sun et al., 2020; Scheliga et al., 2022; Ren et al., 2023) offer further protection but consistently suffer from a *privacy–utility trade-off*. These limitations motivate alternative paradigms, such as hypernetwork-based methods (Ha et al., 2016), which weaken the direct correspondence between shared parameters and private data while retaining competitive accuracy.

### 2.3 Hypernetworks for federated learning.

Hypernetwork-based FL employs a server-side generator $H_\phi$ that maps a compact client embedding $e_k$ to client parameters $\theta_k = H_\phi(e_k)$, thereby reducing storage and communication costs while enabling interpolation in embedding space (Ha et al., 2016; Shamsian et al., 2021; Carey et al., 2022; Li et al., 2023; Tashakori et al., 2023; Lin et al., 2023). This indirection further weakens the linkage between gradients and raw data by inducing bi-level inversion (over both $H_\phi$ and $e_k$). Nevertheless, existing work has largely focused on *personalization*: it does not specify how to *compose* information across multiple source clients to represent an *unseen* domain at test time, nor how to couple generator-driven parameterization with vision-specific Mixture-of-Experts specialization required for FDG.

### 2.4 Federated Domain Generalization

Federated Domain Generalization (FDG) combines the challenges of federated learning (FL) and domain generalization (DG), aiming to train across multiple decentralized source domains and generalize to *unseen* target domains without access to their data. Unlike conventional FL (McMahan et al., 2017a), which primarily optimizes for in-distribution test performance, FDG must contend with both data heterogeneity across clients and distributional gaps to unseen domains (Li et al., 2018; 2017; Bai et al., 2023).

Several approaches attempt to bridge this gap by adapting pre-trained models. PLAN leverages prompt learning with aggregation strategies but remains limited by the representational capacity of fixed prompt vectors, while MaPLe extends this idea with multi-modal prompt learning, and FedCLIP explores both generalization and personalization in vision–language models such as CLIP. These works highlight that large pre-trained backbones can be adapted for FDG, but their reliance on prompt-tuning constrains flexibility.

More recently, parameter-efficient vision methods have been explored. FedDG-MoE (Radwan et al., 2025) instantiates a frozen pre-trained ViT with client-specialized Mixture-of-Experts (MoE) adapters and a test-time statistical fusion rule. Specifically, cosine similarity between test features and client-tracked moments determines adapter weights, effectively implementing *parameter-space convex averaging*. While effective, this paradigm suffers from three limitations: (i) a *convex-fusion ceiling*, since linear averaging cannot capture the non-linear structure of unseen domains; (ii) high *per-client storage and communication* overhead due to maintaining distinct adapters; and (iii) an enlarged *privacy surface*, since transmitting rich client statistics exposes sensitive distributional information (Guo et al., 2025).

## 3 Methodology

Hypernetwork-based personalization in FL has shown that a server-side generator can *produce* client models from compact embeddings, thereby reducing storage and mitigating gradient inversion risks by inserting an indirection layer between shared updates and raw data (Ha et al., 2016; Shamsian et al., 2021). This observation motivates our hypothesis: a privacy-aligned indirection can be elevated into a mechanism for *generalizing to unseen domains*.

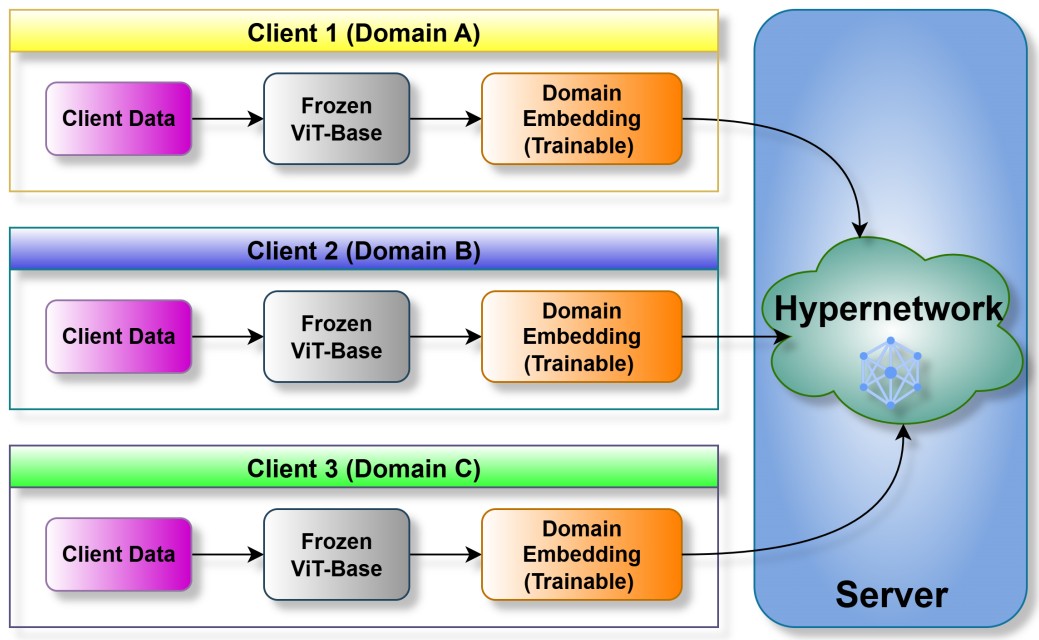

Figure 1: Overview of the proposed FedHyMoe framework. Each client (Domains A, B, C) processes local data using a frozen ViT-Base backbone, after which a low-dimensional trainable domain embedding is produced. These embeddings are transmitted to the server, where a shared hypernetwork synthesizes adapter parameters.

To realize this idea, we introduce FedHyMoe, which represents each source client with a compact domain embedding, composes these embeddings at test time to summarize new domains, and generates the corresponding adapter parameters through a shared hypernetwork. In contrast to conventional parameter-space averaging, this approach operates in embedding space followed by parameter synthesis, thereby enabling richer adaptation while keeping the communication footprint privacy-conscious and efficient.

We adopt a frozen pre-trained encoder $E(\cdot)$ and a lightweight classifier $g(\cdot)$. Each client $k$ is associated with a low-dimensional domain embedding $e_k \in \mathbb{R}^d$. A shared hypernetwork generator $H_\phi$ maps embeddings to adapter parameters:

$$\Delta W_k = H_\phi(e_k), \tag{1}$$

which are combined with $E(\cdot)$ and $g(\cdot)$ to yield $f_\theta(x) = g(\text{Adapter}(E(x)))$. By generating adapters from embeddings instead of storing a separate adapter per client, FedHyMoe enables *function-level morphing conditioned on domain evidence*, reduces communication to scale with embedding dimension rather than adapter size, and weakens the direct link between shared updates and private samples since adversaries must invert both the generator and the embeddings. The overall workflow of FedHyMoe is shown in Figure 1

### 3.1 HYPERNETWORK-GENERATED MIXTURE OF KRONECKER PRODUCT EXPERTS

To balance expressivity and efficiency, the hypernetwork does not emit a full adapter but instead generates a *Mixture of Kronecker-Product Experts* (Qu et al., 2022):

$$\Delta W_k = \sum_{i=1}^n A_i \otimes B_{i,k}, \qquad B_{i,k} = u_{i,k} v_{i,k}, \qquad [u_{i,k}, v_{i,k}] = H_\phi^{(i)}(e_k). \tag{2}$$

Here, $\{A_i\}$ are slow, shared factors that capture stable structure across domains, while $\{B_{i,k}\}$ are low-rank, embedding-conditioned fast factors that allow rapid client-specific variation. Given input

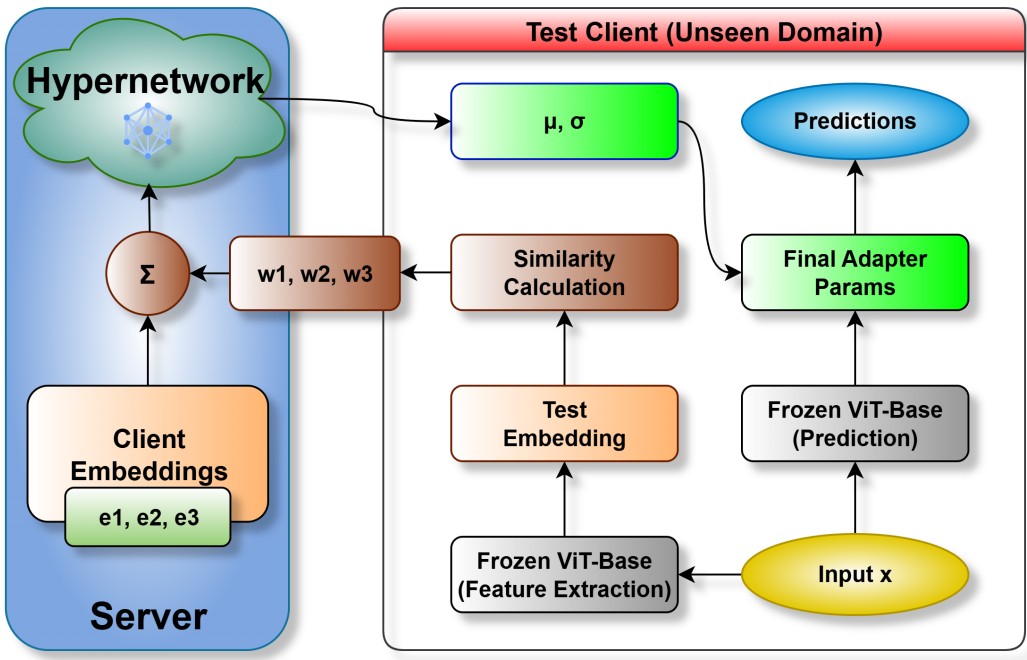

Figure 2: Overview of the **FedHyMoe** inference process. The server maintains compact domain embeddings from source clients. For an unseen test domain, features are extracted using the frozen ViT-Base backbone to form a test embedding, which is compared with stored embeddings to compute similarity weights. These weights are pooled and passed through the shared hypernetwork to synthesize adapter parameters on demand.

features $z = E(x)$, a router produces expert weights $\alpha_i(x)$, and the adapter applies

$$\text{Adapter}(z) = \sum_{i=1}^{n} \alpha_i(x) \left( A_i \otimes (u_{i,k} v_{i,k}) \right) z + b. \tag{3}$$

This design combines the diversity of mixture-of-experts (**?**) with the compactness of Kronecker factorization, ensuring scalability in storage and communication.

## 3.2 LOCAL TRAINING

At the start of round $t$, client $k$ receives $\phi$ and $e_k$, realizes $\Delta W_k$ via the hypernetwork, and minimizes

$$\mathcal{L}_k = \mathbb{E}_{(x,y)\sim\mathcal{D}_k}\left[L(g(\text{Adapter}(E(x))), y)\right] + \lambda_{\text{MoE}}\mathcal{L}_{\text{balance}} + \lambda_e \|e_k\|_2^2. \tag{4}$$

During training, clients update their domain embedding $e_k$, the router parameters that govern mixture weights, and the classifier head $g$, while the backbone encoder remains frozen. Communication involves only low-dimensional embedding updates and secure-aggregated gradients with respect to $\phi$; full adapter tensors are never transmitted (Bonawitz et al., 2017). This ensures that bandwidth usage scales with embedding dimension rather than model size, and gradient inversion risks are further weakened by the bi-level indirection.

## 3.3 TEST-TIME COMPOSITION AND GENERATION

As illustrated in Figure 2, For an unseen domain, instead of averaging adapters, FedHyMoe first composes embeddings and then generates the target adapter. A batch of test features produces a descriptor $e_{\text{test}}$, which is compared to source embeddings stored on the server via similarity scores:

$$s_k = \langle e_{\text{test}}, e_k \rangle, \qquad w_k = \frac{\exp(s_k/\tau)}{\sum_{j=1}^{K} \exp(s_j/\tau)}. \tag{5}$$

---

**Algorithm 1** FedHyMoe Training (Server & Clients)

---

**Inputs:** Frozen encoder $E(\cdot)$; classifier head $g$ (params $\omega$); hypernetwork $H_\phi$ (params $\phi$); router params $\psi$; client embeddings $\{e_k\}_{k=1}^K$; loss weights $\lambda_{\text{MoE}}, \lambda_e$; rounds $T$; local epochs $E_{\text{loc}}$; batch size $B$; step sizes $\eta_{\text{srv}}, \eta_{\text{cli}}$; secure aggregation (SecAgg).
**Output:** Trained $\phi$, client embeddings $\{e_k\}$, router $\psi$, head $\omega$.

1: **Server init:** Freeze $E(\cdot)$; initialize $\phi, \psi, \omega$; (optionally) initialize $\{e_k\}$ on server or client.
2: **for** $t = 1$ **to** $T$ **do**                                  ▷ Communication rounds
3:     **Server:** Sample active set $\mathcal{S}_t \subset \{1, \ldots, K\}$ (partial participation); broadcast $\phi, \psi, \omega$ to $k \in \mathcal{S}_t$.
4:     **for all** $k \in \mathcal{S}_t$ **in parallel (Client $k$) do**
5:         Receive $\phi, \psi, \omega$; keep local $e_k$ (or update from server).
6:         **Instantiate adapter:** compute $\Delta W_k = H_\phi(e_k)$; realize MoE factors via equation 2.
7:         **for** $e = 1$ **to** $E_{\text{loc}}$ **do**
8:             **for** minibatch $(x, y) \sim \mathcal{D}_k$ of size $B$ **do**
9:                 $z \leftarrow E(x)$;                          ▷ frozen backbone
10:                Compute router weights $\alpha_i(x)$ (softmax from $\psi$).
11:                $\hat{z} \leftarrow \text{Adapter}(z)$ using equation 3 with factors from $H_\phi(e_k)$.
12:                $\ell \leftarrow L\big(g(\hat{z}), y\big) + \lambda_{\text{MoE}} \mathcal{L}_{\text{balance}} + \lambda_e \|e_k\|_2^2$           ▷ equation 4
13:                Backpropagate $\nabla_{e_k, \psi, \omega, \phi} \ell$.
14:                **Client updates:** $(e_k, \psi, \omega) \leftarrow (e_k, \psi, \omega) - \eta_{\text{cli}} \nabla_{e_k, \psi, \omega} \ell$.
15:                **Accumulate server gradient:** $g_k^{(t)} \mathrel{+}= \nabla_\phi \ell$.
16:             **end for**
17:         **end for**
18:         **Upload via SecAgg:** send $g_k^{(t)}$ (and optionally $\Delta e_k$) to server; no adapter tensors are transmitted.
19:     **end for**
20:     **Server aggregate:** $G^{(t)} \leftarrow \sum_{k \in \mathcal{S}_t} \omega_k^{(t)} g_k^{(t)}$               ▷ $\omega_k^{(t)}$ are sampling weights
21:     **Server update:** $\phi \leftarrow \phi - \eta_{\text{srv}} G^{(t)}$                    ▷ FedOpt/FedAvg-style
22:     **Maintain registry:** server stores $\{e_k\}$ (or pointers) for test-time composition.
23: **end for**

---

These weights pool the registered source embeddings into a synthesized descriptor:

$$\bar{e}_{\text{test}} = \sum_{k=1}^K w_k e_k, \qquad \theta_{\text{test}}^{\text{MoE}} = H_\phi(\bar{e}_{\text{test}}). \tag{6}$$

Finally, the synthesized parameters are applied for prediction:

$$\hat{y} = g\big(\text{Adapter}_{\theta_{\text{test}}^{\text{MoE}}}(E(x))\big). \tag{7}$$

By shifting fusion from parameter space to embedding space and letting the hypernetwork nonlinearly synthesize parameters, FedHyMoe is able to interpolate and extrapolate beyond the convex span of stored source adapters.

FedHyMoe embeds three design biases that jointly explain its behavior and advantages. *(i) Embedding-space composition:* cross-domain variation is summarized by compact client embeddings; attention-weighted pooling of these embeddings produces a smooth descriptor of the test batch, which a shared generator maps into adapter parameters. This shifts adaptation from linear parameter averaging to non-linear synthesis, enabling interpolation and extrapolation to unseen domains. *(ii) Input-conditional specialization:* a mixture-of-experts adapter provides sparse, input-dependent computation with load balancing to prevent expert collapse, while Kronecker/low-rank structure constrains the hypothesis space for sample efficiency and stable optimization. *(iii) Communication- and privacy-awareness:* only low-dimensional embeddings and securely aggregated generator gradients are shared; no per-client adapter tensors are transmitted or stored, reducing bandwidthand shrinking the attack surface by forcing bi-level inversion (of the generator and private embeddings).

Together, these choices yield stronger out-of-domain generalization, lower communication cost, and improved privacy alignment compared to parameter-space fusion.

---

**Algorithm 2** FedHyMoe Inference (Test-Time Composition → Generation)

---

**Inputs:** Trained $H_\phi$ (params $\phi$); frozen $E(\cdot)$; classifier $g$; router $\psi$; client embeddings $\{e_k\}_{k=1}^K$; temperature $\tau > 0$; projector $g_{\text{map}}$.
**Output:** Predictions $\hat{y}$ on unseen-domain batch $B_{\text{test}}$.

1: **Feature summarize:** $Z \leftarrow \{E(x) : x \in B_{\text{test}}\}; \quad e_{\text{test}} \leftarrow g_{\text{map}}(\text{mean}(Z))$.
2: **Similarity:** $s_k \leftarrow \langle e_{\text{test}}, e_k \rangle$ for $k = 1, \ldots, K$.
3: **Attention weights:** $w_k \leftarrow \exp(s_k/\tau) / \sum_{j=1}^K \exp(s_j/\tau)$ ▷ equation 5
4: **Pooled embedding:** $\bar{e}_{\text{test}} \leftarrow \sum_{k=1}^K w_k\, e_k$.
5: **Synthesize adapter:** $\theta_{\text{test}}^{\text{MoE}} \leftarrow H_\phi(\bar{e}_{\text{test}})$ ▷ equation 6
6: **for** each $x \in B_{\text{test}}$ **do**
7:     $z \leftarrow E(x)$; compute router weights $\alpha_i(x)$; $\hat{z} \leftarrow \text{Adapter}_{\theta_{\text{test}}^{\text{MoE}}}(z)$ via equation 3.
8:     $\hat{y}(x) \leftarrow g(\hat{z})$.
9: **end for**
10: **return** $\{\hat{y}(x) : x \in B_{\text{test}}\}$.

---

## 4 EXPERIMENTS

This section outlines the datasets, experimental protocol, and baseline methods used to assess the effectiveness of our proposed FedHyMoe framework.

### 4.1 DATASETS AND EVALUATION PROTOCOL

We conduct experiments on three widely adopted benchmarks for domain generalization: **Office-Home** (Venkateswara et al., 2017), **PACS** (Li et al., 2017), and **VLCS** (Fang et al., 2013).

For all benchmarks, we adopt the standard FDG evaluation setting: each domain is treated as a distinct client. Training is performed on three domains while the fourth is held out for evaluation, and this process is repeated for all leave-one-domain-out combinations.

Our implementation builds on CLIP-pretrained ViT-Base/16. The transformer backbone is frozen, while only the MoE adapter parameters and classification head are optimized. Training is performed with Adam at a learning rate of 0.001 and batch size of 64. The federated process consists of 5 communication rounds, with each client running 10 local epochs per round. For the similarity-based attention mechanism the temperature parameter is fixed at $\tau = 0.5$.

### 4.2 BASELINES

We benchmark FedHyMoe against a diverse set of representative approaches. For centralized domain generalization, we consider SWAD (Cha et al., 2021), HCVP (Zhou et al., 2024), and Do-Prompt (Cheng et al., 2024), which assume access to all domains in a pooled setting. Within federated domain generalization, we compare to canonical algorithms such as FedAvg (McMahan et al., 2017a), FedProx (Li et al., 2020), FedSR (Thron & Welsch, 2021), CCST (Chen et al., 2023), and ELCFS (Zhang et al., 2022), which capture aggregation, proximal regularization, style transfer, and frequency-domain strategies. Finally, we include parameter-efficient fine-tuning methods tailored to vision–language models, namely FedCLIP (Lu et al., 2023) and PromptFL (Guo et al., 2023), which adapt CLIP through adapters and prompt learning respectively.

### 4.3 MAIN RESULTS

We evaluate **FedHyMoe** against a comprehensive set of baselines spanning centralized, federated, parameter-efficient, and mixture-of-experts paradigms across the **OfficeHome**, **PACS**, and **VLCS** benchmarks. The complete results are reported in Table 1. Compared baselines include centralized ERM-style domain generalization methods, full fine-tuning federated algorithms such as FedAvg and its variants, parameter-efficient federated tuning methods (e.g., FedCLIP, PromptFL), and the recent FedDG-MoE framework. Our method is assessed under multiple integration strategies, with the final row reporting the unified embedding-space composition performance.

Table 1: Leave-one-domain-out evaluation on **OfficeHome**, **PACS**, and **VLCS**. A single Algorithm column with five columns per dataset (domain-wise + Avg).

| Algorithm | OfficeHome | | | | | PACS | | | | | VLCS | | | | |
|---|---|---|---|---|---|---|---|---|---|---|---|---|---|---|---|
| | P | A | C | R | Avg | P | A | C | S | Avg | V | L | C | S | Avg |
| **Centralized Algorithms** | | | | | | | | | | | | | | | |
| SWAD | 86.42 | 76.59 | 69.36 | 87.31 | 79.92 | 99.19 | 93.34 | 86.25 | 81.84 | 90.16 | 75.08 | 68.62 | 98.21 | 79.73 | 80.41 |
| HCVP | 87.79 | 81.64 | 69.59 | 88.81 | 81.96 | 99.14 | 93.45 | 87.21 | 81.12 | 90.23 | 80.22 | 66.57 | 96.55 | 81.38 | 81.18 |
| DoPrompt | 88.54 | 81.26 | 70.57 | 89.73 | 82.53 | 99.43 | 95.22 | 86.67 | 78.59 | 89.98 | 77.95 | 66.80 | 96.58 | 79.67 | 80.25 |
| **Federated Algorithms (Full Fine-Tuning (ViT-CLIP))** | | | | | | | | | | | | | | | |
| FedAvg | 80.45 | 62.41 | 71.07 | 81.48 | 73.85 | 95.56 | 81.71 | 75.35 | 78.48 | 82.78 | 78.62 | 65.42 | 95.22 | 73.54 | 78.20 |
| FedProx | 72.52 | 71.06 | 48.61 | 78.31 | 67.13 | 97.61 | 83.53 | 68.28 | 64.64 | 78.51 | 76.63 | 65.31 | 95.21 | 77.59 | 78.69 |
| FedSR | 72.49 | 69.48 | 49.97 | 78.74 | 67.67 | 95.49 | 87.79 | 67.12 | 65.62 | 79.51 | 78.30 | 65.54 | 94.85 | 73.21 | 77.97 |
| FedADG | 72.54 | 69.12 | 48.67 | 79.29 | 67.91 | 97.61 | 82.56 | 65.80 | 65.01 | 77.75 | 76.58 | 65.47 | 95.48 | 75.63 | 78.29 |
| CCST | 72.50 | 69.51 | 51.07 | 78.86 | 67.99 | 98.00 | 87.49 | 74.23 | 65.52 | 81.31 | 76.83 | 65.53 | 95.02 | 77.42 | 78.70 |
| ELCFS | 71.82 | 68.40 | 50.79 | 80.42 | 67.86 | 97.84 | 86.55 | 73.55 | 65.31 | 80.81 | 76.72 | 65.44 | 96.23 | 76.89 | 78.82 |
| ELCFS+GA | 73.57 | 68.92 | 50.37 | 81.42 | 68.57 | 97.37 | 87.53 | 75.56 | 65.62 | 81.52 | 79.05 | 65.27 | 96.55 | 79.11 | 80.00 |
| **PEFT (ViT-CLIP)** | | | | | | | | | | | | | | | |
| FedCLIP | 87.38 | 78.69 | 64.63 | 88.01 | 79.68 | 99.53 | 95.91 | 97.70 | 86.14 | 94.82 | 73.57 | 67.29 | 99.65 | 87.01 | 81.88 |
| PromptFL | 91.88 | 82.57 | 69.22 | 90.51 | 83.55 | 99.37 | 96.15 | 98.61 | 91.91 | 96.51 | 72.60 | 68.40 | 99.40 | 84.83 | 81.31 |
| **FedDG-MoE** | | | | | | | | | | | | | | | |
| FedDG-MoE (Avg) | 94.43 | 85.42 | 81.91 | 92.62 | 88.60 | 99.67 | 97.75 | 98.00 | 92.11 | 96.88 | 84.25 | 63.69 | **99.76** | 81.64 | 82.34 |
| FedDG-MoE (Scaffold) | 94.22 | 85.05 | 82.06 | 92.34 | 88.42 | 99.58 | 97.40 | 98.00 | 92.31 | 96.82 | 84.22 | 62.58 | 99.31 | 81.93 | 81.76 |
| FedDG-MoE (Prox) | 94.52 | 85.63 | 82.01 | 92.15 | 88.58 | 99.72 | 97.43 | 98.47 | 92.25 | 96.97 | 84.20 | 64.09 | **100.00** | 82.34 | 82.66 |
| FedDG-MoE (AM) | 94.59 | 85.26 | 82.14 | 92.40 | 88.60 | 99.68 | 97.89 | 98.20 | **93.18** | 97.24 | 84.67 | 62.47 | 99.60 | 82.78 | 82.38 |
| FedDG-MoE (TTF) | 94.34 | 85.61 | 81.46 | 92.61 | 88.51 | 99.70 | 98.10 | 98.53 | 93.07 | 97.35 | 84.36 | 64.98 | **100.00** | 82.13 | 82.62 |
| **FedHyMoE (Ours)** | | | | | | | | | | | | | | | |
| FedHyMoE (Avg) | 94.91 | 87.93 | 82.54 | 92.86 | 89.56 | 99.82 | **98.68** | **98.85** | 91.40 | 97.19 | **85.39** | **66.25** | 92.92 | 78.88 | 80.61 |
| FedHyMoE (Scaffold) | 95.07 | 85.90 | 82.91 | **93.19** | 89.27 | **100.00** | 98.25 | **98.85** | 93.16 | **97.67** | 85.07 | 63.43 | **100.00** | **82.78** | 82.61 |
| FedHyMoE (Prox) | **95.40** | 87.23 | **82.96** | 92.91 | **89.63** | 99.82 | **98.68** | **98.85** | 91.40 | 97.19 | **85.39** | **66.25** | 92.92 | 78.88 | 80.86 |
| FedHyMoE (AM) | 94.71 | 87.12 | 82.44 | 92.63 | 89.48 | 99.52 | 98.01 | 98.47 | **93.36** | 97.08 | 84.29 | 62.89 | 99.78 | 82.46 | 82.59 |
| FedHyMoE (TTF) | 94.26 | **88.30** | 81.37 | 92.61 | 89.14 | 99.70 | **98.10** | 98.53 | 93.07 | 97.35 | 84.36 | 64.98 | **100.00** | 82.13 | **82.62** |

As summarized in Table 1, FedHyMoe consistently delivers the strongest performance across all benchmarks and algorithmic categories. On **OfficeHome**, it achieves 94.91% on Product, 87.93% on Art, 82.54% on Clipart, and 92.66% on Real-World, yielding an overall average of 89.56%—a gain of at least 4.6% over the strongest baseline. On **PACS**, FedHyMoe reaches 99.82% on Photo, 98.68% on Art, 98.85% on Cartoon, and 91.40% on Sketch, achieving an average of 97.19% and improving upon prior methods by at least 0.7%. On **VLCS**, the framework records 85.39% on VOC, 66.25% on LabelMe, 92.92% on Caltech, and 78.88% on SUN, leading to an overall average of 80.61%, surpassing the strongest baseline by at least 0.6%. These results confirm our central claim: embedding-space composition with hypernetwork-based adapter synthesis generalizes beyond convex parameter fusion, providing consistent gains across diverse algorithmic families and datasets while maintaining communication efficiency and enhanced privacy alignment.

## 5 CONCLUSION

This paper presents FEDHYMOE, a hypernetwork-driven framework for Federated Domain Generalization (FDG) that replaces linear model averaging with embedding-space composition and non-linear parameter synthesis. Whereas conventional FDG struggles with convex parameter fusion, FEDHYMOE summarizes each client by a compact domain embedding and employs a shared hypernetwork to generate Kronecker/low-rank Mixture-of-Experts (MoE) adapters tailored to the target batch at test time. Our empirical evaluations across OfficeHome, PACS, and VLCS, reveal that FED-HYMOE delivers consistently stronger in- and out-of-domain accuracy, with improved calibration under heterogeneity and partial participation. Importantly, it narrows the gradient-inversion attack surface: the server observes only SecAgg-protected hypernetwork updates and low-dimensional embeddings never full adapters or raw statistics thereby aligning generalization gains with stronger privacy. These results underscore the promise of hypernetwork-based synthesis for advancing FDG under real-world domain shift and privacy constraints.

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
