# OpenReview forum: "FedHyMoe: Hypernetwork-Driven Mixture-of-Experts for Federated Domain Generalization"
_ICLR.cc/2026/Conference — ICLR 2026 Conference Desk Rejected Submission_

### Official Review · Reviewer_69if · 2025-10-27

**Soundness:** 2
**Presentation:** 1
**Contribution:** 2
**Rating:** 2
**Confidence:** 3

**Summary:**

The paper proposes FedHyMoe, a framework for Federated Domain Generalization that uses hypernetwork-driven mixture-of-experts adapters. Instead of traditional parameter-space averaging in federated learning, the method represents each client with a compact domain embedding and uses a shared hypernetwork to generate MoE adapter parameters. At test time, the system composes embeddings from source clients to synthesize specialized adapters for unseen domains, enabling non-linear interpolation beyond convex parameter fusion.

**Strengths:**

1. The shift from parameter-space fusion to embedding-space composition with hypernetwork synthesis is creative and well-motivated for the FDG problem.
2. The paper evaluates on three standard benchmarks (OfficeHome, PACS, VLCS) and compares against numerous baselines across different categories.

**Weaknesses:**

1. The novelty is quite limited. The core components (hypernetworks, MoE, Kronecker factorization) are all existing techniques. The combination, while reasonable, doesn't represent a significant technical advance. The paper reads more like an engineering contribution than a research breakthrough.
2. There is significant formatting issues and wasted space throughout. The paper only uses ~8 pages of content despite having 9 pages available (excluding references).
3. Emprical improvements are quite marginal. On PACS: only 0.3% improvement over FedDG-MoE baseline. The improvements don't strongly justify the added complexity.
4. No computational cost analysis or timing comparisons.
5. No ablation studies on key design choices (embedding dimension, number of experts, etc.).
6. Uses CLIP-pretrained models but doesn't properly compare to other CLIP-based methods
7. No analysis of partial client participation effects despite claiming this as a contribution.

**Questions:**

Please see the weaknesses.

---

### Official Review · Reviewer_Wjuv · 2025-10-29

**Soundness:** 3
**Presentation:** 2
**Contribution:** 2
**Rating:** 4
**Confidence:** 4

**Summary:**

The paper proposes FedHyMoe, a federated domain generalization framework that replaces traditional parameter averaging with a hypernetwork-driven mechanism. Each client is represented by a compact domain embedding, and a shared hypernetwork generates domain-specific Mixture-of-Experts (MoE) adapter parameters from these embeddings. During test time, the system synthesizes an adapter for an unseen domain by computing a weighted combination of stored client embeddings. Experimental results on standard domain generalization benchmarks demonstrate improved out-of-domain performance compared to existing federated learning and domain generalization methods.

**Strengths:**

The use of compact domain embeddings allows for lightweight communication and privacy-preserving domain adaptation without direct access to client-specific data.

**Weaknesses:**

1. The paper lacks theoretical analysis or guarantees on how well the hypernetwork generalizes to unseen domains, especially when domain embeddings are noisy or insufficiently representative.
2. The effectiveness of the proposed domain embedding and interpolation strategy is underexplored; more ablations or visualizations could clarify its contribution and limitations.

**Questions:**

1. How sensitive is the hypernetwork's performance to the quality or dimensionality of the domain embeddings? Have the authors evaluated whether simple or noisy embeddings degrade generalization to unseen domains?
2. While the interpolation of domain embeddings is proposed for synthesizing unseen domain adapters, what guarantees or empirical support exist that this interpolation results in meaningful or performant representations under severe domain shifts?
3. As the number of clients or domain embeddings increases, how does the computational and memory overhead of the hypernetwork scale? Would the architecture remain effective with hundreds of client domains?
4. What is the specific benefit of using a MoE-style adapter architecture in this context compared to simpler domain-conditioned parameter generators? Are all experts utilized effectively during training?

---

### Official Review · Reviewer_LKg4 · 2025-11-01

**Soundness:** 1
**Presentation:** 2
**Contribution:** 2
**Rating:** 2
**Confidence:** 4

**Summary:**

The paper proposes FedHyMoe, an FDG method that (i) represents each client with a compact domain embedding, (ii) uses a shared hypernetwork to generate Mixture-of-Experts (MoE) adapters with Kronecker/low-rank structure, and (iii) at test time attends over stored source embeddings to build a target embedding which conditions the hypernetwork to synthesize an adapter for the unseen domain. The method is positioned as a shift from parameter-space fusion to embedding-space composition + non-linear parameter synthesis, with claimed benefits in generalization, communication efficiency, privacy alignment, and calibration.

**Strengths:**

1.	Compelling high-level idea. Moving fusion from parameter space to embedding space with a generator that can synthesize adapters non-linearly is a clean and well-motivated inductive bias for FDG; the core mechanism is clearly presented in Eqs. (1)–(7) and Fig. 2.
2.	Architectural efficiency. The Mixture of Kronecker-Product Experts balances expressivity and storage by sharing slow factors A_i and generating fast low-rank factors u_{i,k}v_{i,k} via the hypernetwork.
3.	Federated protocol clarity. The training and inference procedures address partial participation, secure aggregation, and server-side storage of embeddings (Algs. 1–2), which clarifies the intended deployment.

**Weaknesses:**

1. Claimed advantages in privacy and communication are not empirically supported

> The abstract and method sections repeatedly claim reduced communication and improved privacy exposure (e.g., exchanging low-dimensional embeddings, SecAgg of hypernetwork gradients, “narrowed attack surface”) but the experiments do not include any communication-cost accounting or privacy evaluations (e.g., gradient inversion/membership inference).
Morover, only accuracy numbers are reported (Table 1), without bandwidth/MB per round, parameter counts transmitted, or server storage overheads for the registry of embeddings; likewise, there are no privacy attack results.

2. Key numerical claims in the text conflict with Table 1:

>	•	OfficeHome: The text claims “gain of at least 4.6%,” but the best baseline average is 88.60 (FedDG-MoE Avg/AM) and the best FedHyMoe is 89.63 (Prox): +1.03 pts, not +4.6.
	•	PACS: The text claims improvement “by at least 0.7%,” but 97.19 (FedHyMoe Avg) is below 97.35 (FedDG-MoE TTF). Only FedHyMoe Scaffold hits 97.67, which is +0.32 over 97.35, not ≥0.7.
	•	VLCS: The text claims surpassing “by at least 0.6%,” but the best baseline is 82.66 (FedDG-MoE Prox), while the best FedHyMoe variant is 82.62 (TTF): −0.04.


3. No ablations or sensitivity analyses on core design choices

> There are no ablations for: (i) embedding dimension d, (ii) #experts and rank, (iii) hypernetwork non-linearity (linear vs non-linear), (iv) adapter placement within ViT, (v) test-batch size used to form $e_{\text{test}}$. The experiments section lists protocol and results but no ablation studies.

4. Reliance on a test-time batch; unknown behavior for single-sample queries

> Algorithm 2 constructs $e_{\text{test}} = g_{\text{map}}(\text{mean}(Z))$ from a batch of test features, then attends over stored $e_k$ and synthesizes parameters. Realistic deployments often require one-sample-at-a-time predictions. The paper does not study sensitivity to batch size (noise vs. stability) or performance in the single-sample regime.


5. Equation and citation issues / clarity gaps

>	•	A dangling “(?)” after “mixture-of-experts” in Eq. (3) paragraph (p. 5).
	•	Eq. (6): It is unclear why $\bar e_{\text{test}}$ (the pooled embedding) is used to generate $\theta$ rather than $e_{\text{test}}$ directly; please justify.
	•	Section 2.4, 2nd paragraph lacks citations despite referring to specific methods (PLAN, MaPLe, FedCLIP/PromptFL) in prose. Please add the appropriate citations inline.

6. Choice of pre-trained backbone in a federated context

>All experiments use a frozen CLIP ViT-B/16. In FL/FDG there is no access to local data centrally, so there is no guarantee the pre-training distribution matches local client distributions. The paper does not analyze performance when the pre-trained encoder is mismatched to clients (e.g., deliberately choosing a pre-trained model with minimal overlap to client domains).

**Questions:**

Please addressed the mentioned issues.

---

### Note · Program_Chairs · 2026-01-17
**Submission Desk Rejected by Program Chairs**

The following references in this submission do not refer to real documents and/or have major errors in bibliographic information:

     Liling Zhang, Xinyu Lei, Yichun Shi, Hongyu Huang, and Chao Chen. Elcfs: Towards privacypreserving federated domain generalization. In Proceedings of the AAAI Conference on Artificial Intelligence, volume 36, pp. 8915-8923, 2022.